# Steroids Static Postural Balance Changes After Exercise Intervention Correlate with Steroidome in Elderly Female

**DOI:** 10.3390/metabo15040239

**Published:** 2025-04-01

**Authors:** Zuzanna Kornatovská, Martin Hill, Dobroslava Jandová, Milada Krejčí, Anna Zwierzchowska

**Affiliations:** 1Department of Education, Faculty of Education, University of South Bohemia in Ceske Budejovice, Jeronymova 10, 37115 Ceske Budejovice, Czech Republic; zkornatovska@pf.jcu.cz; 2Department of Steroid Hormones and Proteofactors, Institute of Endocrinology, 11694 Prague, Czech Republic; mhill@endo.cz; 3Department of Biomedicine, College of Physical Education and Sports PALESTRA, 19700 Prague, Czech Republic; dobroslava.jandova@seznam.cz; 4Department of Wellness and Nutrition Science, College of Physical Education and Sports PALESTRA, 19700 Prague, Czech Republic; krejci@palestra.cz; 5Institute of Sport Science, Academy of Physical Education in Katowice, ul Mikołowska 72a, 40-066 Katowice, Poland

**Keywords:** static postural balance, aging, exercise intervention, steroidome, gas chromatography-tandem mass spectrometry, multivariate statistics

## Abstract

Background: Aging is associated with the development of various disorders, including postural imbalance, which increases the risk of falls and related health complications. This study examines changes in static postural balance after a 4-week intervention involving appropriate exercise and an optimized daily regimen. Additionally, it explores the relationship between these changes and the steroidome. Methods: The study was conducted on a clinically homogeneous group of 41 females around their sixth decade, diagnosed with anxiety-depressive syndrome and treated with selective serotonin reuptake inhibitors (SSRIs). Postural balance was assessed using the dual-scales method (DLLL-DSM), which estimates postural imbalance by evaluating differences in the lower limb load in the standing position. Correlations between initial DLLL-DSM values, age, BMI, and the baseline levels of nine serum steroids, as well as post-intervention changes in five steroids, were analyzed using multivariate regression (OPLS model). Results: A significant reduction in lower limb load differences (-ΔDLLL-DSM), indicating improved postural balance, was observed. The -ΔDLLL-DSM strongly correlated with initial DLLL-DSM values, age, BMI, initial levels of nine serum steroids, and post-intervention changes in five steroids (R = 0.892, *p* < 0.001). Furthermore, initial DLLL-DSM values negatively correlated with adrenal androgen androstenediol sulfate and various sulfated 5α/β-reduced androgen metabolites (R = 0.323, *p* < 0.05), suggesting that the activity of steroid sulfotransferase (SULT2A1) and C17-hydroxylase-C17,20-lyase (CYP17A1) at the lyase step is negatively associated with postural imbalance in elderly females. Conclusions: The findings suggest that even severe postural imbalance can be effectively and relatively rapidly improved through an appropriate exercise-based intervention and an optimized daily regimen, provided that initial adrenal activity is not significantly impaired. Additionally, the identified associations between steroid levels and postural balance provide new insights into the hormonal mechanisms regulating balance control in older individuals.

## 1. Introduction

Coherent and permanent physical activity is considered as an effective instrument for healthy aging and a healthy old age. Movement deficiency in the elderly is characterized by signs summarized in the term “hypokinetic syndrome” manifested as impulsivity, irritability, decreased ability to concentrate as well as an increased anxiety and depression state [1,2,3,4]. Neurocontol of the body posture not only when standing still but also when lying down, and during ambulation, is one of the basic continuous involuntary functions of motor abilities—along with breathing, ideomotoric acts, self-preservation functions, etc. Antigravity postures and positions, especially the bipedal resting upright position, are very challenging to maintain balance with a small base and a highly centered center of gravity of the body and require a continuous interplay of afferent and efferent information with correction. To steroid responses in the context of balance changes in elderly women, some authors [5] concluded that low endogenous serum estradiol levels were associated with greater impairment of postural balance function, whereas hormonal replacement therapy, as compared with placebo, improved postural balance in women with low serum estradiol levels. At the same time, dehydroepiandrosterone (DHEA) metabolites may stimulate aging somatotrophs via estrogen receptors [6]. Moreover, besides the improvement in cognitive abilities, DHEA supplementation promotes activities of daily living [7]. Im et al. [8] found a significant time × group interaction for anterior and posterior dynamic balance, static balance, and growth hormones. After the exercise intervention program, anterior dynamic balance, posterior dynamic balance, static balance, flexibility, muscle strength, growth hormone, dehydroepiandrosterone sulfate (DHEAS), and estrogen significantly increased in the experimental group compared to the control group. The authors concluded that the exercise intervention program contributed to improvements in overall health, including physical function and hormonal status in elderly women. The association of circulating androgens with skeletal muscle mass and function in female seniors was also confirmed by others [9]. While the association between DHEAS and frailty is gender-independent, obesity weakened the relationship between higher DHEA levels and lower frailty status [10]. The use of chairbased exercise programs in institutionalized older women diminished the frailty status and increased the levels of DHEA and testosterone [11]. Bateni [12] showed that physical therapeutic training alone or in addition to Wii Fit (an exergaming video game designed by Nintendo’s Hiroshi Matsunaga) training improved the balance more than Wii Fit training alone.

On the one hand, the circulating levels of both DHEAS and its unconjugated counterpart (DHEA), primarily originating in human adrenal zona reticularis (ZR), are strongly age-dependent and exhibit a prominent decrease in seniors of both genders [13]. On the other hand, an appropriate physical exercise and modification of the daily regimen can achieve at least partial restoration of ZR functioning and reinstatement of DHEAS/DHEA levels as demonstrated in our previous studies [14,15].

ZR is the deepest zone of the human adrenal cortex and its occurrence in mammals is species-specific. Whereas Cytochrome B5 (CYB5) is extensively expressed in the ZR which induces the preference of the lyase step in C17-hydroxylase-C17,20-lyase (CYP17A1) enzyme over the C17 hydroxylase. CYB5 activity also inhibits the expression of type 2 3β-hydroxysteroid dehydrogenase (HSD3B2) in ZR but promotes the expressions of steroid sulfotransferase SULT2A1 and family 1 type 3 aldoketoreductase (AKR1C3) participating in the formation of extragonadal testosterone. Therefore, the DHEAS is the primary product of the ZR and is known as the most biologically active steroid in humans acting in a non-genomic way [16,17]. Even if the ZR activity is stimulated by ACTH, the response of steroidogenic enzymes in the ZR to ACTH is weaker than in zona fasciculata (ZF). Moreover, CRH primarily stimulating the pituitary secretion of ACTH may also directly stimulate ZR activity [18,19,20]. The aging adrenal cortex in both sexes shows a declining production of adrenal androgens, a decline in CYB5 activity, lessening activity of the lyase step in CYP17A1, and a diminished ZR mass [21].

The authors of this study draw on the concept that uncertainty associated with postural balance disorders translates into the personality characteristics of older people and is associated with insecurity even at the mental level and vice versa, imbalances in the emotional sphere; deepening anxiety to depressive symptoms lead to imbalances at the somatic level. Older people often suffer from emotional lability, affective disorders, insecurity or loss of ”balance” in interaction with external stressors, exacerbating the hypokinetic syndrome, anxiety and fears of falls and conditions deepening into depression and postural insecurity. In our previous studies, we focused on the effects of physical exercises (pre/post a 4-week exercise intervention) and an improvement in the daily regimen in Priessnitz Spa Jeseník on the functioning hypothalamic-pituitary-adrenal axis and by extension on the mental balance using a steroidomic approach (pre/post changes in the steroidome). Here, we hypothesize that there is also an improvement in the static postural balance using the dual-scales method after the intervention (-ΔDLLL-DSM) and that this improvement is related to ΔDLLL-DSM and both starting levels as post-intervention changes in the steroidome.

## 2. Materials and Methods

### 2.1. Participants

The research was focused on examinations of a clinically homogeneous group of 41 female postmenopausal volunteers in the age 58 (55, 61) years (shown as the median with quartiles). The following exclusion criteria were applied. Persons with acute or more severe chronic neurological or orthopedic disease, which would make it difficult or significantly modify the examination of standing on two scales. Persons with an analysis of biomechanical causes of postural imbalance, such as congenital (shorter lower limbs, asymmetry of femoral neck angles, etc.), were not included in the study. Exclusion from the study also occurred in subjects after total endoprosthesis and in others with limited proprioceptive information and a large change in ambulation or an inability to stand without support. Also, persons with severe proprioception in a severe diabetic peripheral polyneuropathy, in severe residual neurological findings after central nervous system inflammation, and all persons with contraindications valid in the field of medical rehabilitation and balneology determined according to the White Book on Physical and Rehabilitation Medicine in Europe [22] were excluded from the study. It provided a basic hematological and biochemical screening to exclude acute contraindications and for a steroid metabolome at the pre/post examination.

All participants were on medication with selective serotonin reuptake inhibitors. The patients underwent a standardized 1-month intervention therapy with physical activity and an optimized daily regimen in a spa in the Czech Republic. All of the participants signed up for the research voluntarily, which they also confirmed by signing the document in the medical record. An instructional interview was given to all participants of the study regarding the dual-scale method measurement, the movement intervention, and blood collection for examination. Informed consent was obtained from all the patients included in the study. The Ethical Committee of the research institution expressed full agreement with the research aims and procedures conforming with the requirements stipulated in the Declaration of Helsinki. Ethic Committee Name: Ethic Committee of the College of Physical Education and Sports PALESTRA, Prague, Czech Republic: Approval Code: VŠP/0699/2022: Approval Date: 16 February 2022.

### 2.2. Medical History

A standardized medical history protocol was applied to assess the health state, psychic state, disabilities, operations, etc., of each participant, and finally, the physician carried out the medical recommendation/decision to include an individual in the study.

### 2.3. Assessment of Body Composition

Body height was measured using the Tanita Leicester Height Measure device (Invicta Plastics, Leicester, UK) with an accuracy of 0.1 cm. A tetrapolar multi-frequency bioelectrical impedance device InBody 230 (InBody, Seoul, Republic of Korea) was used to assess body weight, BMI, body fat percentage, and total muscle mass [4].

### 2.4. Postural Balance Examination

A postural balance examination was conducted by a specialized rehabilitation physician. The “dual-scale method”, measuring left/right weight-bearing using two identically calibrated weighing scales, was performed on SENCOR SBS 113 LS digital scales with a load capacity of up to 150 kg, placed in a way that they touched the base and the LCD displays were on the sides. The participant stepped on the scales as closely as possible to the “middle line” with each lower limb on one scale in a symmetrical position of the middle line soles of the feet. Left/right weight-bearing measured using two scales is a consistent method for evaluating weight distribution through the legs. The short- and long-term weight-bearing tendencies showed a similar degree of variation. Weight-bearing inequalities were not associated with any significant left/right differences in bone mineral density at the hip, but were weakly associated with left-right differences in leg muscle mass [23] (Figure 1). The measuring was repeated a total of three times: the first time, participant looked straight ahead at a point on the wall at eye level; the second time, the participant walked around the scales and stood on them staring into the distance out of the window; the third time, the participant walked through the surgery for about 5 m with one turn back and forth and stood on the scales again with their eyes fixed on the distance. After that, an algorithm recording all three values was used [23,24].

### 2.5. Blood Sampling

The participants underwent the blood sampling pre/post the 4-week exercise intervention. At 7 a.m., 10 mL of blood was drawn from the cubital or forearm vein (following overnight fasting) into a cooled plastic tube containing 100 µL of 5% EDTA. Plasma was obtained by centrifugation for 5 min at 2000× *g* at 4 °C, separated and frozen within half an hour of being drawn from the subject, and stored at −20 °C until analyzed. The participants were instructed not to eat nuts, bananas, or tomatoes, and not to drink coffee or black tea for 2 days before sampling [14].

### 2.6. Intervention

The 4-week exercise intervention on weekdays took place during the spa stay (Priessnitz Spa in Jeseník) and consisted of simple and basic yoga exercise lessons and walking sets with sticks (with an entrance test of ascent and descent on calibrated wooden stairs) [25]. The yoga lessons were focused on body posture and balance control, flexibility, muscle strength, breathing, the stimulation of psychic harmony, and the optimization of social interaction [4,26]. The exercises were carried out in accordance with the system of Yoga in Daily Life [26], and without contraindications to the elderly (Sarvahita Asanas), whilst sitting on a chair or standing. Once a week, the main training lesson, lasting 90 min, was conducted in small groups of 10–12 participants under the guidance of the coach, and with two or three coach assistants who helped participants to complete exercises easily and correctly. After the main training lesson, each participant received an educational sheet, which contained a simple guide and a symbolic attribute for the concrete intervention week. During the week, participants repeated practiced exercises each day for 20–30-min. The assistants also motivated participants during these weeks to repeat the learnt exercises. The week program also included the mottoes: Week 1, “You are never alone”; Week 2, “Change is always possible”; Week 3, “Movement is life”; Week 4, “Enjoy life and every moment” [27,28].

Nordic walking sets always started with warming-up exercises and breathing preparation, followed by walking uphill in a field and then walking on the flat terrain of the spa route in the forest park at a medium pace (100–110 TF/min) within 25 min. The last part of the Nordic walking sets consisted of the immersion of limbs according to the typical Priessnitz spa method.

The study was conducted according to the guidelines of the Declaration of Helsinki, and approved by the Institutional Ethics Committee of the College of Physical Education and Sports PALESTRA (protocol code: GACR_01_2016; date: 17 March 2016 and protocol code VŠP_0699_2022; date: 16 February 2022).

### 2.7. Statistical Analysis

The variables (including their post-intervention changes) entering the statistical analysis were as follows: postural balance, age (initial values only), height (initial values only), body mass index (BMI), and 71 steroids and steroid conjugates. The differences in postural balance changes and changes in steroid levels pre/post intervention exercise were analyzed by a robust Wilcoxon’s paired test using NCSS 12 statistical software (version 12) from a Number Cruncher Statistical System (Kaysville, UT, USA).

To estimate the associations between post-intervention changes in postural balance (dependent variable) on one hand and steroidome and its post-intervention changes on the other (predictors), we used multivariate regression with a reduction in dimensionality known as the model of orthogonal projections to latent structures (OPLS) [28,29,30,31]. The same model was used to evaluate the associations between postural balance and steroidome before intervention.

The OPLS model simultaneously evaluates the relationships between dependent variables and predictors and is capable of coping with the problem of severe multicollinearity (high intercorrelations) in the matrix of predictors, while ordinary multiple regression fails to evaluate such data. The multicollinearity in OPLS is favorable as it enhances the predictivity of the model. The variability shared between dependent variables and predictors was explained by one predictive component while orthogonal components explained the variability shared within the highly intercorrelated predictors independently of the dependent variable. Although necessary for OPLS model building, these orthogonal components were not interesting from the interpretation point of view.

Orthogonal projections to latent structures identified the best predictors as well as the best combination of predictors to estimate postural balance. After standardization of the variables, the OPLS model can be expressed as follows:X = T P + T P + E T T p p 0 0 (1)Y = T P + F T p p (2)
where X is the matrix with predictors and subjects and Y is the vector of dependent variables and subjects; T p is the vector of component scores from the single predictive component and subjects extracted from Y; To is the vector of component scores from the single orthogonal component and subjects extracted from X; Pp is the vector of component loadings for the predictive component extracted from Y; Po is the vector of component loadings for the orthogonal component extracted from X and independent variables; and E and F are error terms.

The relevant predictors were chosen using variable importance of the projection (VIP) statistics. The statistical software used for OPLS analysis (SIMCA-P v.12.0 developed by Umetrics AB, Umeå, Sweden) enabled us to determine the number of relevant components, detect multivariate non-homogeneities, and test multivariate normal distribution and homoscedasticity (constant variance). The algorithm for obtaining the predictions is shown elsewhere [32].

### 2.8. Steroid Analysis

Steroids were determined in blood samples collected for analysis blood sample collection in the amount of 5 mL, always in the morning, in a sitting position, in the Endocrinology Institute in Prague. The collection was performed by a nurse under the supervision of a physician with subsequent processing of the sample in the laboratory. The GC-MS/MS method quantification of steroids was described in detail in our previous article [33]. To assess the statistical power, we conducted a power analysis and calculated the effect size for key variables. The effect size was estimated using Cliff’s delta, and the power analysis was performed using the Wilcoxon test.

Effect sizes for selected variables: DLLL-DSM: −0.25 (moderate effect size); BMI: −0.25 (moderate effect size); Pregnenolone: 0.0 (no significant difference); Cortisol: 0.0 (no significant difference); Corticosterone: 0.0 (no significant difference).

Power analysis: DLLL-DSM: 0.025 (low power); BMI: 0.025 (low power); Pregnenolone: 0.026 (low power); Cortisol: 0.026 (low power); Corticosterone: 0.026 (low power).

The current sample size (n = 41) results in low statistical power, which limits the ability to detect significant differences. To achieve the recommended power of 0.8, the sample size would need to be approximately 1571 participants, which is impractical due to logistical and practical constraints in conducting clinical research.

Improving statistical power in the current study without increasing the sample size

Given the constraints in recruiting a larger sample, we applied more efficient statistical methods to enhance the power of tests in this study.

Mixed ModelsWe accounted for between-subject variability, which allowed for a more precise estimation of the intervention effect.The mixed model demonstrated significant changes in DLLL-DSM and BMI, although the limited sample size affected the precision of estimates.Bayesian Analysis

We estimated mean changes and 95% confidence intervals for key variables: DLLL-DSM: Mean change = −7.53, 95% CI: (−7.53, −7.53); BMI: Mean change = −2.68, 95% CI: (unstable values); Pregnenolone: Mean change = −0.0894, 95% CI: (unstable values); Cortisol: Mean change = −29.2, 95% CI: (unstable values); Corticosterone: Mean change = −1.91, 95% CI: (unstable values). Bayesian analysis enables a better interpretation of results in small samples.

### 2.9. Terminology of Steroid Polar Conjugates

Concerning the terminology of the steroid polar conjugates used here, the term steroid sulfate was used in the case of the dominance of 3α/β-monosulfate over other forms of steroid conjugates, while the term conjugated steroid was used in the case of comparable amounts of conjugate forms (sulfates, disulfates, and glucuronides). This terminology was based on the relevant literature, with appropriate citations for each steroid as follows: Pregnenolone sulfate [34,35], 20α-dihydropregnenolone sulfate, dehydroepiandrosterone (DHEA) sulfate [35,36,37], androstenediol sulfate [35,36], allopregnanolone sulfate, isopregnanolone sulfate [38], conjugated pregnanolone (sulfate + glucuronide) [39], conjugated 5α-pregnane-3β,20α-diol (3β,20α-disulfate + 3β-sulfate) [39], conjugated 5β-pregnane-3α,20α-diol (3β,20α-disulfate + 3β-sulfate + glucuronide) [39], androsterone sulfate [35,36], epiandrosterone sulfate [35,36], etiocholanolone sulfate [40], epietiocholanolone sulfate, conjugated 5α-androstane-3α,17β-diol (glucuronide + sulfate) [36], and conjugated.

## 3. Results

All the evaluated parameters which significantly changed after the intervention are shown in Table 1. The most important parameter DLLL-DSM showed a significant postintervention reduction as did the values of BMI. Out of the 18 steroids showing significant change, 15 rose and only 3 significantly declined. The circulating levels of most steroids in the Δ5 and Δ4 steroidogenic pathways increased after the intervention as did some of their reduced metabolites. Furthermore, the levels of two glucocorticoids (cortisol and corticosterone) and 11β-hydroxyandrostenedione levels also significantly increased.

The relations between basal DLLL-DSM and steroids are displayed in Table 2. DLLL-DSM is negatively associated with an adrenal androgen androstenediol sulfate and various sulfated 5α/β-reduced androgen metabolites. However, this association is moderate as only 10.5% of variability in DLLL-DSM is explained by steroid levels.

Alternatively, there is a strong association between the decrease in DLLL-DSM (-ΔDLLL-DSM, improvement in postural balance) vs. DLLL-DSM, initial levels of steroids and steroid post-intervention changes as 79.7% of variability in -ΔDLLL-DSM is explained by these predictors in the OPLS model (Table 3). As is obvious, age and the values of BMI, DLLL-DSM and levels of nine steroids before intervention positively correlate with -ΔDLLLDSM. In addition, a decrease in levels of four other steroids significantly positively correlates with -ΔDLLL-DSM, while the change in epietiocholanolone sulfate showed a negative correlation. Of the steroids before intervention, whose levels significantly positively correlated with the -ΔDLLL-DSM, three were the Δ5 steroids (pregnenolone, 17-hydroxypregnenolone sulfate, and 7-oxodehydroepiandrosterone), one was the Δ4 steroid (androstenedione), four were 5α/β-reduced metabolites of progesterone and 17-hydroxyprogesterone (isopregnanolone sulfate, conjugated 5β-pregnane-3α,20α-diol, 5αpregnane-3α,17,20α-triol, 5α-pregnane-3β,17,20α-triol) and one was a reduced 11 β-hydroxy androgen (11βhydroxyepiandrosterone). Of the steroid post-intervention changes, which significantly positively correlate with the -ΔDLLL-DSM in the OPLS model, two involve the Δ5 steroids (20α-dihydropregnenolone sulfate, DHEA), one the Δ4 steroid (16α-hydroxyprogesterone), and one the 17-deoxycorticosteroid (corticosterone). Only the sulfated 5β-reduced metabolite of DHEA epietiocholanolone after intervention exhibits negative correlation with the -ΔDLLL-DSM. The multiple regression (MR) model in Table 3 illustrates severe multicollinearity among the steroids as the steroids which are significant in the OPLS model lose significance in the MR model. Only preintervention levels of etiocholanolone reach a significance in the MR model, which indicates that the steroid significantly correlates with -ΔDLLL-DSM independently of the correlations with the remaining predictors. The same applies to age, BMI, and initial DLLL-DSM, which is the most important predictor of -ΔDLLL-DSM.

## 4. Discussion

Postural control is essential for active daily activities and represents an integral part of independence and mobility in the elderly. Successful postural control in elderly females depends on individual assumptions (motor skills, sensors, and cognition), interaction of information from the environment, and especially on provided simple physical exercises performed regularly and adequately according to one’s individual potential. All these factors should be taken into account in the individual and differentiated therapy of female instability in presenium and senium when a standard of standing on two scales is a recognized difference of up to 6 kg, or up to 10% of weight for overweight people over 100 kg [24].

This study is in accordance with others showing positive associations between levels of DHEA and its metabolites on one hand and postural stability on the other [5,6,7,8,9,10]. In accordance with other authors [41,42], we can affirm that acquired negative postural habits should be compensated by regular stretching of shortened muscle groups and we can conclude that regular intervention based on walking, breathing gymnastics, and compensatory exercises may optimize the functioning of the adrenal cortex and recover the postural stability during a relatively short period of 4 weeks.

The significant post-intervention changes in levels of steroids and overall improvement in adrenal functioning are for the most part in accordance with the data of our previous studies [14,15], in which we detected significant changes in 29 steroids and discussed them in detail in the context of their biological effects. In this study, we detected significant changes in 18 steroids. Some discrepancy concerning the number of significantly changed steroid levels when comparing the present study with our previous ones may be due to the smaller number of volunteers included in this study (n = 41 vs. n = 45) and consequently the reduced power of statistical analysis and/or due to the use of absolute changes instead of relative differences, which were used in our previous studies. In this study, the absolute differences were used to diminish the effects of steroid levels before intervention on post-intervention changes because evaluating the correlation of post-intervention steroid changes with post-intervention changes in postural stability was the primary objective of the study.

Since the significance of post-intervention changes in steroids and their associations with improvements in neurotic scores was discussed in detail in our previous study [14,15], we primarily discussed here a post-intervention improvement in postural stability in relation to both steroid starting levels and their post-intervention changes. This improvement was highly significant (Table 1), which confirms our hypothesis that physical exercises (pre/post 4-week exercise intervention) and improvement in the daily regimen in Priessnitz Spa Jeseník result in a significant improvement in postural balance, which is closely associated with a concomitant improvement in the adrenal steroidogenesis as with the functioning of the adrenal cortex at the beginning of the intervention (Table 3). These changes are in line with the data from our previous studies, which were focused on post-intervention improvement in neurotic scores [14,15].

The data in Table 2 and Table 3 demonstrate an association of steroids with postural stability (both steroid starting levels vs. postural stability and steroid post-intervention changes and starting levels vs. post-intervention changes in postural stability). These relations include the steroids that directly arise in the adrenal cortex but also include their metabolites. The greater the increase in most steroids is (see also Table 1), the smaller is the post-intervention improvement in postural balance. As in the case of a decrease in neurotic scores, which was discussed in our previous study (see [14]), these findings might also be linked to high levels of CRH. CRH is a stress marker stimulating the synthesis of C21 steroids in the adrenal zona fasciculata (ZF) and C19 steroids (androgens) in the zona reticularis (ZR) via the stimulation of ACTH secretion in the pituitary and consequent activation of ZF and ZR, but also by the direct stimulation of androgen production in the zona reticularis (ZR) (see [14]). Therefore, the increased concentrations of C21 and C19 steroids after the intervention may also reflect the partial persistence of stress, and consequently a less efficient effect of the intervention on the decline in DLLL-DSM. Besides the relations with pre/post-exercise steroid levels, the improvement in postural balance strongly depends on the starting values of DLLL-DSM. The more pronounced the starting postural imbalance is, the more pronounced is its post-intervention recovery. These results also show similarities to some data of our previous study, namely the relations between improvements in somatic symptoms on the one hand and steroids before intervention and their changes after intervention on the other [14].

Concerning more specific relations of steroidomes at the beginning of the intervention and its post-intervention changes on one hand and post-intervention changes in postural balance on the other, the post-intervention fall in the DLLLDSM positively correlates with pre-intervention levels of pregnenolone, 17-hydroxypregnenolone sulfate (both originate in adrenal ZF), and their catabolites such as isopregnanolone sulfate, conjugated 5β-pregnane-3α,20αdiol, unconjugated 5α/β-pregnane-3α/β,17,20α-triols, and also with DHEA metabolites 7-oxo-DHEA, androstenedione, etiocholanolone and finally with 11β-hydroxy-androstanes, which may originate via the CYP11B1 catalyzed conversion of 11-deoxy-androstanes to their 11β-hydroxy-counterparts and/or via cleavage of corticoids catalyzed by CYP17A1 in the C17,20-lyase step [43]. With regard to the bioactivity of the steroids listed above, unconjugated pregnenolone functions as an inhibitory neurosteroid on low conductivity voltage-gated calcium channels (L-type VGCC) but operates as a positive modulator on type 3 melastatin receptors (TRPM3) and binds to pregnane-X type receptors (PXR) participating in the elimination of xenobiotics and endogenous toxic substances as do also the 17-hydroxypregnenolone, androstenedione, and etiocholanolone. Moreover, the activation of PXR may also attenuate some inflammatory processes (see [14]). Isopregnanolone sulfate as well as its unconjugated counterpart negatively modulate inhibitory type A GABA receptors (GABAAR) (see [14]). Alternatively, the unconjugated 5β-pregnane-3α,20α-diol is a positive modulator of inhibitory GABAAR and functions as an ergosteroid activating the enzymes glycerol-3-phosphate dehydrogenase and malic enzyme and inducing increased heat production. 7-oxo-DHEA effects the same exertion (see [14]) and also inhibits the type 1 11β-hydroxysteroid dehydrogenase (HSD11B1) converting an inactive cortisone to glucocorticoid cortisol, the excess of which is diabetogenic. The etiocholanolone is an additional ergosteroid and operates also as a positive modulator of inhibitory GABAAR and at the same time as a negative modulator of the vanilloid receptor (TRPV1) mediating pain transmission at the peripheral level.

With regard to the significant post-intervention increase in corticoids, a positive relationship between one of them and the post-intervention improvement in postural balance (Table 3), the explanation is the same as for the post-intervention improvement in mental balance, which was thoroughly discussed in our previous study (see [14]). In brief, aging is associated with declined negative feedback on the cortisol secretion due to impaired sensitivity of the hypothalamo-pituitary-adrenal axis (HPAA). The elevation of cortisol levels is frequently related to a lower cognitive status, dementia, anxiety, depression, and neurodegenerative diseases. In addition, the risk of developing insulin resistance increases not only with higher cortisol levels in older seniors but also with the flatter diurnal slope of cortisol in these subjects, which is associated with type 2 diabetes. A potential ambiguity concerning the slight but significant post-intervention increase in glucocorticoids such as cortisol, corticosterone, and 11β-hydroxyandrostenedione, which is either their metabolite or a substance whose formation is catalyzed by the same enzyme type 1 11 β-hydroxylase (CYP11B1) as well as a positive relationship between corticosterone and an improvement in postural balance in the OPLS model may be explained by a shift in higher corticosteroid levels from evening and night to morning, when samples were taken, though this is the desirable adjustment in the cortisol diurnal profile.

Concerning the more specific relationships of steroidome and postural balance at the beginning of the intervention, there are inverse relations between levels of several adrenal androgens including their reduced metabolites and DLLL-DSM (Table 2), which points to a plausible association of ZR activity with postural balance as both CYP17A1 in the lyase step converting C21 steroids to their C19 counterparts and sulfotransferase SULT2A1 are the key enzymes forming the conjugated adrenal androgens, whose levels negatively correlate with increasing age. As regards the absence of DHEA sulfate as a relevant steroid in the corresponding OPLS model but the relevance of androstenediol sulfate, this phenomenon may be linked to the activity of the important enzyme functioning in ZR type 3 subfamily 1C aldoketoreductase (AKR1C3), which may convert DHEA to androstendiol [44]. From the steroids correlating with pre-intervention postural stability, the unconjugated counterpart of androstenediol sulfate is a potent immunostimulant [45] (see also [46]). Epipregnanolone sulfate is a negative modulator of excitatory L-type VGCC. The sulfates of androsterone and epiandrosterone negatively modulate neuroinhibitory GABAAR and glycine receptors (GlyR). Furthermore, the epiandrosterone sulfate positively modulates TRPM3 receptors and thus stimulates the insulin synthesis in pancreatic β-cells.

### Limitations

The purpose of the presented study was fulfilled. We are aware of some limitations of the study. The physical activity level of participants and story of falls were not evaluated before the intervention. Despite all efforts to follow the basic research methodology, the examined sample could not fully reflect the general population of female elderlies in their sixth decade. A larger sample size would be beneficial for future research.

The current sample size (N = 41) results in low statistical power, which limits the ability to detect significant differences. To achieve the recommended power of 0.8, the sample size would need to be approximately 1571 participants, which is impractical due to logistical and practical constraints in conducting clinical research.

We recommend further research to confirm the results. This study can use a single-blind design to avoid potential bias. We would like to emphasize that follow-up studies should monitor the sustainability and stability of the results after 1–3–6 months following the intervention.

## 5. Conclusions

This study shows that positive adjustment of the postural balance, like an adjustment of mental imbalance in older women, should be investigated in a biological and hormonal context where the reduction in postural imbalance through appropriate physical training and adjustment of the daily regimen is closely associated with a relatively rapid improvement in the adrenal cortex function. Bayesian analysis enables a better interpretation of results in small samples, and in our study, it confirmed the significant impact of the intervention on DLLL-DSM and BMI. The positive associations between the initial values of postural imbalance and levels of favorable steroids on one hand and post-intervention improvement in postural balance on the other hand, indicate that even a serious postural imbalance may be efficiently treated by the aforementioned intervention provided that the initial adrenal activity is not seriously disrupted. The results confirm significant changes in DLLL-DSM and BMI, but they also highlight the need for cautious interpretation of other variables due to the limited statistical power. However, further studies are required to elucidate the corresponding mechanisms of action.

## Figures and Tables

**Figure 1 metabolites-15-00239-f001:**
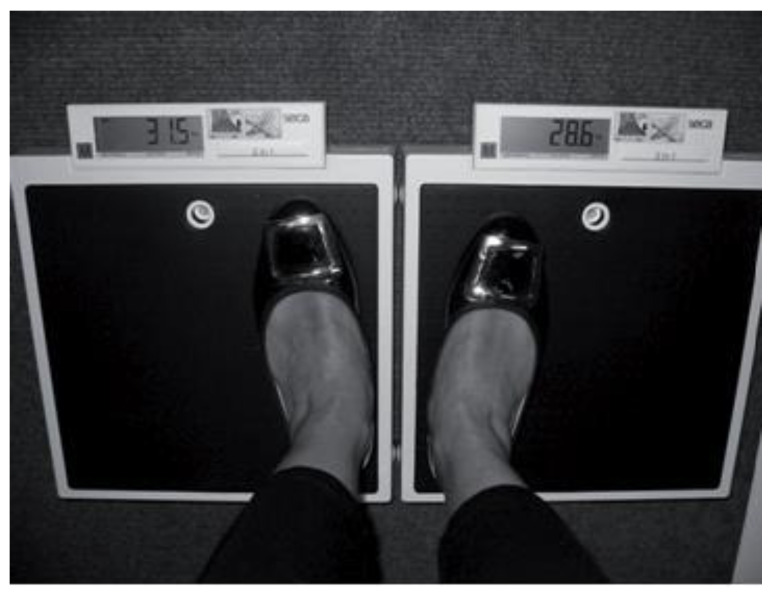
Participant standing astride two identical scales in a natural standing posture [24].

**Table 1 metabolites-15-00239-t001:** Changes in steroids and further parameters after intervention by physical activity during the spa stay (shown as median with quartiles) in 41 female patients. Except for age and height, only parameters with significant (*p* < 0.05) changes are shown.

Variable	Initial Values	Post-Intervention Changes	*p*-Value ^b^
DLLL-DSM ^a^ [% of body weight]	7.69 (4.26, 13)	−2.56 (−7.53, 0)	<0.001
Age [years]	58 (55, 61)	-----	-----
Height [cm]	164 (160, 170)	-----	-----
BMI [kg/m^2^], BMI change [%]	25.5 (23.7, 31.2)	−1.2 (−2.68, 0)	0.001
Pregnenolone [nM]	0.935 (0.717, 1.18)	0.207 (−0.0894, 0.426)	0.017
Pregnenolone sulfate [nM]	79.6 (53.1, 106)	7.29 (−7.94, 18.6)	0.028
17-Hydroxypregnenolone [nM]	1.62 (1.09, 3.02)	0.47 (−0.087, 3.07)	0.010
17-Hydroxypregnenolone sulfate [nM]	3.21 (2.18, 5.68)	0.925 (−0.36, 2.9)	0.020
Dehydroepiandrosterone (DHEA) [nM]	6.75 (4.3, 8)	1.78 (−0.306, 3.71)	0.007
Androstenediol [nM]	1.29 (1.02, 1.97)	0.139 (−0.1, 0.39)	0.045
17-Hydroxyprogesterone [nM]	0.77 (0.502, 1.22)	0.175 (−0.209, 0.882)	0.034
16α-Hydroxyprogesterone [nM]	0.458 (0.287, 1.01)	0.196 (−0.125, 0.51)	0.039
Androstenedione [nM]	3.11 (1.98, 4.16)	0.608 (−0.405, 2.01)	0.034
Conjugated epipregnanolone [nM]	1.56 (0.837, 2.43)	0.14 (−0.165, 0.445)	0.027
Conjugated 5α-pregnane-3α,20α-diol [nM]	4.69 (3.43, 8.02)	−0.366 (−1.4, 0.372)	0.009
5α-Pregnane-3α,17α,20α-triol [nM]	0.201 (0.122, 0.385)	−0.0485 (−0.131, 0.0307)	0.014
5β-Pregnane-3α,17α-20α-triol [nM]	2.41 (1.05, 4.63)	−0.159 (−1.43, 0.152)	0.049
Epietiocholanolone sulfate [nM]	43.6 (23.5, 65.3)	4.49 (−0.694, 14.2)	0.008
Conjugated 5β-androstane-3α,17β-diol [nM]	0.721 (0.389, 1.14)	0.139 (−0.0561, 0.326)	0.012
Cortisol [nM]	413 (291, 472)	84.6 (−29.2, 189)	0.029
Corticosterone [nM]	12.6 (6.72, 17.5)	4.26 (−1.91, 13.7)	0.006
11β-Hydroxyandrostenedione [nM]	136 (84, 201)	8.79 (−11.8, 48.6)	0.031

^a^ Differences between loads on the lower limbs while standing on 2 scales (DBLLL), ^b^ The significance of changes was evaluated using a Wilcoxon’s robust paired test.

**Table 2 metabolites-15-00239-t002:** Relations between basal DLLL-DSM and steroids as evaluated by OPLS and multiple regression models (for details, see Statistical analysis).

	Predictive Component OPLS				Multiple Regression		
Variable	Component Loading	*t*-Statistic	R ^a^		Regression Coefficient	*t*-Statistic	
Androstenediol sulfate	−0.301	−13.23	−0.805	**	−0.033	−2.73	*
Conjugated epipregnanolone	−0.266	−7.39	−0.711	**	−0.033	−1.92	*
Androsterone sulfate	−0.297	−5.37	−0.796	**	−0.029	−1.48	
Epiandrosterone sulfate	−0.331	−12.05	−0.888	**	−0.035	−2.08	*
Etiocholanolone sulfate	−0.305	−6.50	−0.816	**	−0.029	−1.84	
Epietiocholanolone sulfate	−0.319	−20.95	−0.854	**	−0.033	−1.98	*
Conjugated 5α-androstane-3α,17β-diol	−0.308	−5.25	−0.824	**	−0.045	−2.68	*
Conjugated 5α-androstane-3β,17β-diol	−0.325	−10.72	−0.870	**	−0.036	−1.73	
Conjugated 5β-androstane-3α,17β-diol	−0.330	−10.10	−0.884	**	−0.045	−2.41	*
Conjugated 5β-androstane-3β,17β-diol	−0.270	−5.75	−0.724	**	−0.041	−1.97	*
11β-Hydroxyandrosterone sulfate	−0.298	−7.93	−0.798	**	−0.037	−2.59	*
DLLL-DSM ^b^	1.000	2.21	0.323	*			
Explained variability			10.5% (5.6% after cross-validation)				

^a^ R—Component loadings expressed as correlation coefficients with predictive component, * *p* < 0.05, ** *p* < 0.01, ^b^ DLLL-DSM—differences between load on the lower limbs while standing on 2 scales (% of body weight).

**Table 3 metabolites-15-00239-t003:** Relations DLLL-DSM vs. DLLL-DSM, initial levels of steroids and steroid post-intervention changes as evaluated by OPLS and multiple regression models (for details, see Statistical analysis).

	Variable	Component Loading	*t*-Statistic	R ^b^		Regression Coefficient	*t*-Statistic	
Relevant predictors (matrix X)	Age	0.246	2.30	0.352	*	0.313	4.34	**
	BMI	0.205	2.56	0.292	*	0.186	2.04	*
	DLLL-DSM ^a^	0.538	3.04	0.767	**	0.665	10.81	**
	Pregnenolone	0.166	4.40	0.237	**	0.096	1.51	
	17-Hydroxypregnenolone sulfate	0.169	2.13	0.241	*	0.010	0.11	
	7-oxo-Dehydroepiandrosterone	0.118	2.10	0.168	*	0.027	0.27	
	Androstenedione	0.186	2.57	0.265	*	0.076	0.73	
	Isopregnanolone sulfate	0.152	2.25	0.216	*	−0.003	−0.04	
	Conjugated 5β-pregnane-3α,20α-diol	0.182	2.15	0.260	*	0.034	0.43	
	5α-Pregnane-3α,17,20α-triol	0.230	2.05	0.328	*	−0.019	−0.41	
	5α-Pregnane-3β,17,20α-triol	0.232	1.91	0.331	*	0.071	0.63	
	5β-Pregnane-3α,17,20α-triol	0.184	1.83	0.262		0.058	0.89	
	Etiocholanolone	0.220	1.46	0.315		0.210	1.97	*
	11β-Hydroxyandrosterone	0.145	1.73	0.207		−0.054	−0.56	
	11β-Hydroxyepiandrosterone	0.195	2.37	0.278	*	0.030	0.42	
	11β-Hydroxyetiocholanolone	0.166	1.65	0.237		0.026	0.57	
	Δ20α-Dihydropregnenolone sulfate	−0.157	−2.35	−0.224	*	−0.109	−1.67	
	ΔDehydroepiandrosterone (DHEA)	−0.119	−2.77	−0.171	*	0.082	0.90	
	Δ16α-Hydroxyprogesterone	−0.137	−3.07	−0.195	**	0.005	0.04	
	ΔEpiandrosterone	−0.132	−2.14	−0.188	*	0.091	1.05	
	ΔEpietiocholanolone sulfate	0.208	1.88	0.298		0.124	1.90	*
	ΔCorticosterone	−0.196	−5.56	−0.279	**	−0.039	−0.69	
(matrix Y)	ΔDLLL-DSM	−1.000	−9.45	−0.910	**			
Explained variability		79.7% (60.9% after cross-validation)						

^a^ DBLLL-DSM—differences between load on the lower limbs while standing on 2 scales (% of body weight) ^b^ R—Component loadings expressed as correlation coefficients with predictive component, * *p* < 0.05, ** *p* < 0.01, Δ—post-intervention change.

## Data Availability

The datasets analyzed during the current study are not publicly available. Data sharing not applicable due to the General Data Protection Regulation (GDPR), legislatively valid protection of personal data in the European Union, valid in the Czech Republic since 25 May 2018.

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
