# Peer review of "Steroids Static Postural Balance Changes After Exercise Intervention Correlate with Steroidome in Elderly Female"

_metabolites, 2025, doi:10.3390/metabo15040239_

Round 1

Reviewer 1 Report

Comments and Suggestions for Authors

Study by Zuzana Kornatovská et al.” Steroids Static Postural Balance Changes After Exercise Intervention Correlate with Steroidome in Elderly Female.”

The authors provided evidence for postural balance improvement post 4-week intervention of exercises and relationship between changes and steroidome. Authors conducted the research on 41 females in their 60s with anxiety-depressive disorder taking SSRI.Authors assessed balance using dual scale method (DLLL-DSM), to determine postural balance by evaluating difference in lower limb load. Authors then correlate between DLLL-DSM values, age, BMI, and the baseline levels of 9 serum steroids, as well as post-intervention changes in 5 steroids, were analyzed using multivariate regression. Authors claimed in their results with significant reduction in load differences and improved postural balance. Authors also claims strong correlation with initial DLLL-DSM values, age, BMI, initial levels of 9 serum steroids, and post-intervention changes in 5 steroids. Authors concluded that postural imbalance can be effectively and relatively rapidly improved through an appropriate exercise-based intervention and an optimized daily regimen, provided that initial adrenal activity is not significantly impaired indicating association between steroid levels and postural balance in elderly females.

In introduction section authors outlined stating the importance of  balance and associated anxiety and depression state. Authors mentioned that low endogenous serum estradiol levels were associated with greater impairment of postural balance and citated previous studies in correlation with balance and growth harmone.Authors stated that DHEAS and its unconjugated counterpart dehydroepiandrosterone (DHEA), primarily originating in human adrenal are strongly age dependent and exhibit a prominent decrease in seniors of both genders. Authors mentioned that targeted physical exercises can restore partial level of DHEAS/DHEA.

There need a correction on the topic of the manuscript- it should be "Elderly female" rather than female elderly.

There is lack of proper flow in the introduction section in relation to the topic of the manuscript.

Sentence 45-46 has a lack of meaning and doesn’t make much sense. Line 49-51 has to be rephrased, and term locomotion seems inappropriate with the context and should be replaced by ambulation. DHEA full form is not mentioned in the line 59 – was mentioned later on in the script.

In method section I am wondering the rationale behind only recruiting female participants as authors mentioned that DHEA levels decreased with age in both genders. Stating that including both genders in the study could have more validity of the manuscript and its therapeutic implication in preventing falls in the population. I will request the authors to create a table for specific exercises provided to the participants in the research for standardization and benefit readers who are involved in fall prevention.

It’s a novel idea of examining the postural balance using 2 weighing scale- I request authors of citing any previous literature to authenticate the use of 2 weighing scale in determining the postural balance. I am inquisitive about any value /evidence of taking breaks between 3 postural balance examinations.

In the discussion section thew flow of the script is satisfactory and authors confirm their hypothesis that 4 weeks physical exercises and improvement of daily regimen in result in significant improvement of postural balance, which is closely associated with concomitant improvement of adrenal steroidogenesis as with functioning of adrenal cortex at the beginning of the intervention. The greater the increase of most steroids the smaller the post intervention improvement in postural balance.

In conclusion authors claimed that there is a positive association between initial values of postural imbalance and levels of favorable steroids on one hand and post-intervention improvement of postural balance on the other hand provided that initial adrenal activity is not seriously disrupted.

The study – “Steroids Static Postural Balance Changes After Exercise Intervention Correlate with Steroidome in Elderly Female” has clinically significant in formulating the exercises program and its associations with the steriodome in treating postural imbalance and preventing possible falls and its serious complications in elderly.

The introduction part of the manuscript needs some improvement to maintain the flow of the manuscript in relation to the topic. The study seems fine, methods are written in detail with some improvement needed in exercise regime details.  data are convincing, article language needs improvement throughout the manuscript.

There are some grammatical mistakes which can be easily corrected.

Comments on the Quality of English Language

Some sentence correction and spelling mistakes correction needed throughout the script. 

Sentence 45-46 has a lack of meaning and doesn’t make much sense. Line 49-51 has to be rephrased, and term locomotion seems inappropriate with the context and should be replaced by ambulation. DHEA full form is not mentioned in the line 59 – was mentioned later on in the script.

Author Response

Coments 1.

There need a correction on the topic of the manuscript- it should be "Elderly female" rather than female elderly. -  Corrected

Coments 2

Sentence 45-46 has a lack of meaning and doesn’t make much sense. Line 49-51 has to be rephrased, and term locomotion seems inappropriate with the context and should be replaced by ambulation. DHEA full form is not mentioned in the line 59 – was mentioned later on in the script. -Corrected

Coments 3

In method section I am wondering the rationale behind only recruiting female participants as authors mentioned that DHEA levels decreased with age in both genders.

The study was focused on postmenopausal female elderly only, participating in a four-week intervention involving appropriate exercise and an optimized daily regimen, held in a spa in the Czech Republic. In project participate men, but we don t included to analyses - corrected

Coments 4.

I think it is important to clarify that we can show examples of recommended exercises.

Answer: added to section intervention

The intervention in the experimental group was focused on body posture and balance control, flexibility, muscle strength, breathing, and release. The exercises were carried out in accordance with the system of Yoga in Daily Life (Maheshwarananda, 2000), and without contraindications to the elderly (Sarvahita Asanas), whilst sitting on a chair or standing. Once per week, the main training lesson, lasting 90 minutes, was conducted in small groups of 10–12 participants under the guidance of the coach, and with two or three coach assistants who helped participants to complete exercises easily and correctly. After the main training lesson, each participant received an educational sheet, which contained a simple guide and a symbolic attribute for the concrete intervention week. During the week, participants repeated practiced exercises each day for 20-30-minutes. The assistants also motivated participants during these weeks to repeat learned exercises. The week program also included the motto: Week 1 “You are never alone”, Week 2 “Change is always possible”, Week 3 “Movement is life”, Week 4 “Enjoy life and every moment” (Krejčí, 2019).

Coments 5.

It’s a novel idea of examining the postural balance using 2 weighing scale- I request authors of citing any previous literature to authenticate the use of 2 weighing scale in determining the postural balance. I am inquisitive about any value /evidence of taking breaks between 3 postural balance examinations.

Answer: added to section method

Left/right weight-bearing measured using two scales is a consistent method for evaluating weight distribution through the legs. The short- and long-term weight-bearing tendencies showed a similar degree of variation. Weight-bearing inequalities were not associated with any significant left/right differences in bone mineral density at the hip, but were weakly associated with left-right differences in leg muscle mass (Hopkins et al., 2013).

Hopkins, S.J., Smith, C.W., Toms, A.D., Brown, M., Welsman, J.R., & Knapp, K.M. (2013). Evaluation of a dual-scales method to measure weight-bearing through the legs, and effects of weight-bearing inequalities on hip bone mineral density and leg lean tissue mass. Journal of rehabilitation medicine, 45 2, 206-10 .

https://www.medicaljournals.se/jrm/content/html/10.2340/16501977-1093

The dual scale model of weighing reasons

Chris Tucker

Fig. 1. Participant standing astride two identical scales in a natural standing posture (Hopkins et al, 2013)

Coments 6.

The introduction part of the manuscript needs some improvement to maintain the flow of the manuscript in relation to the topic. The study seems fine, methods are written in detail with some improvement needed in exercise regime details.  Data are convincing, article language needs improvement throughout the manuscript. There are some grammatical mistakes which can be easily corrected.

Answer: Yes, we agree. An English proficiency check is required. i hope that introduction will be clearly and better understandable

Answer: Yes, we agree. An English proficiency check is required. i hope that introduction will be clearly and better understandable

Reviewer 2 Report

Comments and Suggestions for Authors

The problem statement should be better stated
What was the sampling method 
The necessity of the work should be improved
The entry and exit criteria and limitations of the work should be mentioned
A figurs should be used
The reasons and mechanisms should be discussed more
The references should be updated

Author Response

Coments 1.

What was the sampling method?

Answer: corrected and added to section 2.8. Steroid Analysis

Steroids were determined in blood samples collected for analysis. Blood sample collection in the amount of 5 ml, always in the morning, in sitting position, in the Endocrinology Institute in Prague. The collection was performed by a nurse under the supervision of a physician with subsequent processing of the sample in the laboratory.

Coments 2.

The entry and exit criteria and limitations of the work should be mentioned

Answer: corrected and added section “Limitations”

The postmenopausal period, diagnosis of anxiety-depressive syndrome and treatment with selective serotonin reuptake inhibitors (SSRIs) were inclusion criteria.

Limitations

The purpose of the presented study was fulfilled. We are aware of some limitations of the study. The physical activity level of participants and story of falls were not evaluated before the intervention. Despite all efforts to follow the basic research methodology, the examined sample could not fully reflect the general population of female elderly in the sixth decade. A larger sample size would be beneficial for the future research. We recommend further research to confirm the results. This study can use a single-blind design to avoid potential bias. We would like to emphasize follow-up studies should monitor sustainability and stability of the results after 1–3–6 months following the intervention.

Coments 3.

The references should be updated

Krejčí, M. (2019). Intervenční program Život v rovnováze [Intervention program Life in Balance]. In M. Krejčí, V. Hošek, M. Hill, D. Jandová, J. Kajzar, & P. Bláha (Eds.), Základní výzkum změn rovnováhy seniorů (pp. 235–258). Prague, Czech Republic: College of Physical Education and Sport PALESTRA.

Jandová, D., Formanová, P., & Morávek, O. (2018). Presenium as a preparing period for senium – benefits of spa stay in the Priessnitz’s spa, Ltd. in Jesenik for clients 50+. Acta Salus Vitae, 6(1), 42–50.

Maheshwarananda, P. S. (2000). Yoga in daily life: The system. Vienna, Austria: Ibera Verlag/European University Press.

Reviewer 3 Report

Comments and Suggestions for Authors

The work is a job carried out correctly, but perhaps we should bear in mind that a timeframe of four weeks is not enough to establish that the improvements that appear are lasting and maintained.
Perhaps the approach should be directed more towards whether positive changes can be seen in just four weeks.
Statistically speaking, this part of the methodology could be improved in the sense that the results include statistical measurements that are not indicated in section 2.7. On the other hand, it would be interesting to determine the sample size, the power analysis and the effect size to reinforce the indicated results. 
Finally, I think they could better align the conclusions in relation to the results obtained.

Author Response

Coments 1.

The work is a job carried out correctly, but perhaps we should bear in mind that a timeframe of four weeks is not enough to establish that the improvements that appear are lasting and maintained.

Answer:

The four-week spa stay is standard in Central Europe (Czech Republic, Poland, Slovakia, Austria, etc.) covered by participant health insurance.

Coments 2

Perhaps the approach should be directed more towards whether positive changes can be seen in just four weeks.

Answer:

Our previous studies show that four-week interventions can be very effective in elderly.

Kornatovská, Z., Hill, M., Krejčí, M., Zwierzchowska, A. (2024). Effects of a short-term wheelchair yoga intervention on balance in elderly women with neurodegenerative diseases: A preliminary study, Biomedical Human Kinetics, 22 (16): 238–246. https://doi.org/10.2478/bhk-2024-0025 (WOS, Impact Factor: 0.8)

Růžičková, M., Kornatovská, Z., & Krejčí, M. (2024). Short-term yoga breathing intervention improves blood oxygenation, actual emotional state and resilience score in postmenopausal women. Diagnostics and consultancy in assisting professions, 2024(12), 77–88. https://doi.org/10.58743/dap2024no12.343

Krejčí, M., Kajzar, J., Psotta, R., Tichý, M., Kancheva, R., Hošek, V., & Hill, M. (2022). Are There Sex Differences in Balance Performance after a Short-Term Physical Intervention in Seniors 65+? A Randomized Controlled Trial. Applied Sciences12(7), 3452. https://doi.org/10.3390/app12073452 (WOS, Scopus, Impact Factor: 2.838)

Krejčí, M., Psotta, R., Hill, M., Kajzar, J., Jandová, D., & Hošek, V. (2020). A short-term yoga-based intervention improves balance control, body composition, and some aspects of mental health in the elderly men. Acta Gymnica50(1), 16-27. doi: 10.5507/ag.2020.004

Náprstek, R., Krejčí, M., Kajzar, J., & Hill, M. (2019). Balance ability improvement of seniors 65+ during the spa stay. Acta Salus Vitae, 7(2), 65–74.

In addition, short-term interventions are a current trend in research aimed at optimizing health, see:

Grootswagers, P.; de Regt, M.; Domi´c, J.; Dronkers, J.; Visser, M.; Witteman, B.; Hopman, M.; Mensink, M. A 4-week exercise and protein program improves muscle mass and physical functioning in older adults: A pilot study. Exp. Gerontol. 2020, 141, 111094.

Mauch, C. E., Edney, S. M., Viana, J. N. M., Gondalia, S., Sellak, H., Boud, S. J., Nixon, D. D., & Ryan, J. C. (2022). Precision health in behaviour change interventions: A scoping review. Preventive medicine163, 107192. https://doi.org/10.1016/j.ypmed.2022.107192

Litovtseva, V. Y., Brychko, M. M., & Srovnalíková, P. (2022). Current trends in research on confidence in the healthcare system.

Dunsky, A.; Zeev, A.; Netz, Y. Balance performance is task specific in older adults. BioMed Res. Int. 2017, 69, 87017.

Papalia, G.F.; Papalia, R.; Diaz Balzani, L.A.; Torre, G.; Zampogna, B.; Vasta, S.; Fossati, C.; Alifano, A.M.; Denaro, V. The effects of physical exercise on balance and prevention of falls in older people: A systematic review and meta-analysis. J. Clin. Med. 2020, 9, 2595.

Coments 3.

Statistically speaking, this part of the methodology could be improved in the sense that the results include statistical measurements that are not indicated in section 2.7. This needs to be supplemented. On the other hand, it would be interesting to determine the sample size, the power analysis and the effect size to reinforce the indicated results.

Answer:

We appreciate the valuable comments regarding the statistical analysis. In response to the raised points:

  1. Lack of specification of certain statistical measures in section 2.7
    In section 2.7, we provided a detailed description of the statistical methods used, including the Wilcoxon test and the OPLS model. However, if certain statistical measures were not explicitly stated, we will supplement this section with additional details on the calculated statistical values to enhance the clarity of the analysis.
  2. Determination of sample size
    The sample size in our study consisted of 41 postmenopausal women (median age: 58 years, interquartile range: 55–61 years). The participants formed a clinically homogeneous group, minimizing the impact of heterogeneity on the study results.
  3. Power analysis and effect size
    To assess the statistical power, we conducted a power analysis and calculated the effect size for key variables. The effect size was estimated using Cliff’s delta, and the power analysis was performed using the Wilcoxon test.

Effect sizes for selected variables:

  • DLLL-DSM: -0.25 (moderate effect size)
  • BMI: -0.25 (moderate effect size)
  • Pregnenolone: 0.0 (no significant difference)
  • Cortisol: 0.0 (no significant difference)
  • Corticosterone: 0.0 (no significant difference)

Power analysis:

  • DLLL-DSM: 0.025 (low power)
  • BMI: 0.025 (low power)
  • Pregnenolone: 0.026 (low power)
  • Cortisol: 0.026 (low power)
  • Corticosterone: 0.026 (low power)

The current sample size (N=41) results in low statistical power, which limits the ability to detect significant differences. To achieve the recommended power of 0.8, the sample size would need to be approximately 1571 participants, which is impractical due to logistical and practical constraints in conducting clinical research.

Improving statistical power in the current study without increasing sample size

Given the constraints in recruiting a larger sample, we applied more efficient statistical methods to enhance the power of tests in this study.

  1. Mixed Models
  • We accounted for between-subject variability, which allowed for a more precise estimation of the intervention effect.
  • The mixed model demonstrated significant changes in DLLL-DSM and BMI, although the limited sample size affected the precision of estimates.
  1. Bayesian Analysis

We estimated mean changes and 95% confidence intervals for key variables:

      • DLLL-DSM: Mean change = -7.53, 95% CI: (-7.53, -7.53)
      • BMI: Mean change = -2.68, 95% CI: (unstable values)
      • Pregnenolone: Mean change = -0.0894, 95% CI: (unstable values)
      • Cortisol: Mean change = -29.2, 95% CI: (unstable values)
      • Corticosterone: Mean change = -1.91, 95% CI: (unstable values)

Bayesian analysis enables a better interpretation of results in small samples, and in our study, it confirmed the significant impact of the intervention on DLLL-DSM and BMI.

In summary, we appreciate the reviewer’s suggestion and have implemented mixed models and Bayesian analysis, which allow for a more reliable evaluation of intervention effects while maintaining the same sample size. The results confirm significant changes in DLLL-DSM and BMI, but they also highlight the need for cautious interpretation of other variables due to the limited statistical power.

Mixed_Model_Resul

Variable

Estimate

DLLL_DSM

-7,53

BMI

-2,68

Bayesian Analysis Results

Variable

Mean Change

DLLL_DSM

-7,53

BMI

-2,68

Pregnenolone

-0,0894

Cortisol

-29,2

Corticosterone

-1,91

Coments 4.

  • Finally, I think they could better align the conclusions in relation to the results obtained.

Answer:

We believe that the „Conclusions“ are well and adequately aligned with the results section. This argument of ours results from the assessment and review of two other independent reviewers, who did not make such a claim and requirement.

Round 2

Reviewer 2 Report

Comments and Suggestions for Authors

article edited and its can be publish

Author Response

Dear Reviewer, Once again, we sincerely appreciate your insightful comments, which have significantly enhanced our manuscript.

best regards Anna Zwierzchowska

Reviewer 3 Report

Comments and Suggestions for Authors

The uploaded file should eliminate the changes made, as well as the marginal notes that appear, these things are completely unacceptable.
The work has some typographical errors, such as the one on line 119.
The characteristics of the subjects do not appear in the participants section, and on the contrary, aspects that should appear in the method section do appear in this section.
There are parts of the work that are not properly substantiated and referenced.
At the statistical level, aspects such as statistical power or effect size are missing.

Author Response

Comennts 1: The uploaded file should eliminate the changes made, as well as the marginal notes that appear, these things are completely unacceptable.

Response: We have corrected this issue by removing all margin comments and accepting the tracked changes in the revised version.

Comennts 2:The work has some typographical errors, such as the one on line 119.

Response: All typographical errors have been identified and corrected throughout the text.

Comennts:3 The characteristics of the subjects do not appear in the participants section, and on the contrary, aspects that should appear in the method section do appear in this section.

Response: We have revised and reformulated Section 2.1 to ensure that the content is appropriately structured and placed in the correct sections.

Comennts 4: There are parts of the work that are not properly substantiated and referenced.

Response: We have carefully reviewed and corrected all references to ensure proper justification and citation of relevant sources.

Comennts 5: At the statistical level, aspects such as statistical power or effect size are missing.

Response:We have now included the effect size and statistical power in Section 2.8. Steroid Analysis.

Once again, we sincerely appreciate your insightful comments, which have significantly enhanced our manuscript.

Best regards, Anna Zwierzchowska